# A Mild Presentation of X-Linked Hypophosphatemia Caused by a Non-Canonical Splice Site Variant in the *PHEX* Gene

**DOI:** 10.3390/genes15060679

**Published:** 2024-05-24

**Authors:** Gloria Fraga, M. Alba Herreros, Marc Pybus, Miriam Aza-Carmona, Melissa Pilco-Teran, Mónica Furlano, M. José García-Borau, Roser Torra, Elisabet Ars

**Affiliations:** 1Pediatric Nephrology Department, Hospital de la Santa Creu i Sant Pau, Institut de Recerca Sant Pau (IR Sant Pau), RICORS-SAMID, Universitat Autònoma de Barcelona, 08193 Barcelona, Catalonia, Spain; gfraga@santpau.cat; 2Nephrology Department, Fundació Puigvert, Institut de Recerca Sant Pau (IR-Sant Pau), RICORS2040 (Kidney Disease), Department of Medicine, Universitat Autònoma de Barcelona, 08193 Barcelona, Catalonia, Spain; 3Molecular Biology Laboratory, Fundació Puigvert, Institut de Recerca Sant Pau (IR-Sant Pau), RICORS2040 (Kidney Disease), 08193 Barcelona, Catalonia, Spain; 4Neonatology Unit, Pediatrics Department, Hospital de la Santa Creu i Sant Pau, Universitat Autònoma de Barcelona, 08193 Barcelona, Catalonia, Spain

**Keywords:** X-linked hypophosphatemia, *PHEX* gene, non-canonical splice site variant

## Abstract

X-linked hypophosphatemia (XLH) is a rare inherited disorder of renal phosphate wasting with a highly variable phenotype caused by loss-of-function variants in the *PHEX* gene. The diagnosis of individuals with mild phenotypes can be challenging and often delayed. Here, we describe a three-generation family with a very mild clinical presentation of XLH. The diagnosis was unexpectedly found in a 39-year-old woman who was referred for genetic testing due to an unclear childhood diagnosis of a tubulopathy. Genetic testing performed by next-generation sequencing using a kidney disease gene panel identified a novel non-canonical splice site variant in the *PHEX* gene. Segregation analysis detected that the consultand’s father, who presented with hypophosphatemia and decreased tubular phosphate reabsorption, and the consultand’s son also carried this variant. RNA studies demonstrated that the non-canonical splice site variant partially altered the splicing of the *PHEX* gene, as both wild-type and aberrant splicing transcripts were detected in the two male members with only one copy of the *PHEX* gene. In conclusion, this case contributes to the understanding of the relationship between splicing variants and the variable expressivity of XLH disease. The mild phenotype of this family can be explained by the coexistence of *PHEX* transcripts with aberrant and wild-type splicing.

## 1. Introduction

X-linked hypophosphatemia (XLH) (OMIM 307800; ORPHA 89936) is the most common form of inherited phosphate wasting and has an estimated prevalence of between 1.7 per 100,000 children and 4.8 per 100,000 individuals [1,2]. XLH is caused by loss-of-function variants in the phosphate-regulating endopeptidase homolog X-linked (*PHEX)* gene located on Xp22.1, which is inherited with an X-linked dominant pattern. Pathogenic variants in the *PHEX* gene cause an increase in the serum levels of fibroblast growth factor-23 (FGF23), which decrease the renal tubular reabsorption of phosphate as well as reducing the synthesis of 1,25-dihydroxyvitamin D [1,25(OH)_2_D], resulting in chronic hypophosphatemia [3].

Patients with XLH usually present with a clinical phenotype consisting of rickets in children, osteomalacia in adults, short stature, lower extremity deformities, and dental anomalies. These conditions cause pain, stiffness, and a decrease in physical activity, which in turn affect quality of life [1,2]. However, XLH can manifest with a broad phenotypic spectrum ranging from mild isolated hypophosphatemia to severe lower extremity bowing.

XLH frequently becomes apparent within the first two years of life, when lower extremity bowing becomes noticeable with the onset of weight bearing. The diagnosis of XLH is frequently delayed because some patients present with mild manifestations; however, early diagnosis is crucial as appropriate treatment could prevent the complications associated with the disorder. Diagnosis is based on clinical, radiological, and biochemical findings, but genetic testing is considered the gold standard to confirm the disease [1]. In this study, we show the genotype–phenotype correlation of a non-canonical intronic *PHEX* variant associated with a very mild form of XLH in a three-generation family.

## 2. Materials and Methods

A three-generation family was clinically and genetically studied due to suspected renal tubulopathy. Written informed consent was obtained for genetic testing. Next-generation sequencing (NGS) and Sanger sequencing were performed as part of routine diagnostic activities. All studies were conducted in accordance with the principles of the Helsinki declaration on medical research involving human subjects.

DNA and total RNA were extracted from peripheral blood samples from the 3 family members using the Wizard Genomic DNA purification Kit (Promega, Madison, WI, USA) and the TRIzol^®^ reagent (Invitrogen, Waltham, MA, USA), respectively, according to the manufacturer’s instructions. Genetic testing of the consultand was performed by the NGS of a custom-designed targeted kidney disease gene panel including 316 genes causative of or associated with inherited kidney diseases (including the *PHEX* gene), as previously described [4]. Splicing effect prediction was performed with the deep learning-based splice variant software SpliceAI (version 1.3.1) and SQUIRLS (version 2.0.1) using the default threshold values [5,6]. Segregation analysis was performed by PCR and Sanger sequencing of the exon 18 and its flanking intron boundaries of the *PHEX* gene in the consultand’s father and son.

RNA characterization of the non-canonical splice site variant identified in the *PHEX* gene was performed by reverse transcription (RT)–PCR with Sanger sequencing of PCR products. RT (1 µg of RNA) was prepared using the High-Capacity Reverse Transcription Kit (Applied Biosystems, Foster City, CA, USA). PCR amplification and Sanger sequencing of the *PHEX* transcripts were performed using the following primers: Forward 5′-GAGTTATGGTGCTATAGGAG-3′ (located on exon 17) and Reverse 5′-GTCTGTAGGAATTGCACCTC-3′ (located on exon 21).

## 3. Results

### 3.1. Clinical Description

The consultand was a 39-year-old woman referred to the nephrology unit due to an unclear diagnosis of renal tubular acidosis at 3 months of age in her country of origin. She had been treated with bicarbonate for 6 years during her childhood. She was concerned about this uncertain diagnosis because she had become a mother. At the time of consultation, she had no clinical or laboratory findings suggestive of acidosis, hypokalemia, hypercalciuria, or hypocitraturia, and kidney function was normal with a creatinine level of 61 μmol/L (estimated glomerular filtration rate [eGFR] 109 mL/min/1.73 m^2^ CKD-EPI). She presented normal levels of phosphorus, magnesium, calcium, and parathyroid hormone (PTH); no glycosuria or aminoaciduria; and renal tubular reabsorption of phosphate (TRP) of 82%. Only low levels of serum 25OHD (33 nmol/L [normal range > 50 nmol/L) were detected (Table 1). There were no signs of nephrocalcinosis or lithiasis on ultrasound. The height of the patient was 1.66 m.

Her father, a 70-year-old man, reported non-specific muscle and bone pains since the age of 55, which have remained stable and occasionally been treated with paracetamol. His height was 1.70 m, and he had few bone defects, such as minor ankle varus and flat foot, bilateral coxarthrosis with total hip prosthesis at age 67, and slight pectus excavatum. He was regularly physically active. Blood tests showed normal kidney function, calcium and alkaline phosphatase, decreased phosphate (0.69 mmol/L), high PTH (9 pmol/L), and low 1,25(OH)_2_D (12 pg/mL). He had been prescribed calcitriol (0.25 mg/day) and phosphate (3.5 g/day). Phosphate levels normalized after starting treatment. Urine analysis showed a low TRP of 68%, with no other findings of proximal tubulopathy (Table 1).

Her son, a 3-year-old boy, presented mild hypophosphatemia; other values were normal (Table 1). The patient did not have limb deformity and his growth velocity was normal. At 2 years and 6 months, his anthropometrics were as follows: weight, 20 kg (>p99; 3.56 standard deviation [SD]); height, 1.03 m (>p99; 3.25 SD); body surface area, 0.76 m^2^, body mass index, 18.85 kg/m^2^ (p99; 2.19 SD). His kidneys were normal on ultrasound, he had no glycosuria or hyperuricosuria, and he had a normal excretion profile of amino acids in urine.

### 3.2. Genetic Study

The genetic study of the consultand was performed by the NGS of a broad kidney disease gene panel and identified an intronic variant in the *PHEX* gene (NM_000444.6): c.1900-4_1900-1dup in a heterozygous state. No pathogenic variants were detected in the other 59 tubulopathy-associated genes included in the gene panel [4]. This *PHEX* variant consisted of a duplication of 4 intronic nucleotides (TTAG) within the region of the splice acceptor of intron 18, including the canonical -1,-2 acceptor splice site in addition to -3,-4 nucleotides upstream from the exon 19 boundary (Figure 1). The variant was absent in the Genome Aggregation Database (gnomAD v.4.0) in a well-covered region and was not reported in either the literature or the ClinVar database. Two computational algorithms predicted a deleterious effect on the splicing of this variant with the following maximum scores: SpliceAI Score = 1.0; SQUIRLS Score = 1.0. Segregation analysis of the variant showed that her father and her son both carried this *PHEX* variant in hemizygosity. RNA analysis by RT-PCR and Sanger sequencing of the resulting amplicons confirmed aberrant splicing at the novel splice site, resulting in the inclusion of the four intronic nucleotides in the coding sequence. Expression of the aberrantly spliced and wild-type transcripts was detected in both males (Figure 1). This variant was predicted to generate a change in the protein translation reading frame, resulting in a truncated protein. According to the ACMG criteria [7], the variant was classified as likely pathogenic (PS3, PM2_supporting, PP3).

## 4. Discussion

The availability of NGS technologies has enormously facilitated the etiologic diagnosis of families with atypical presentations of monogenic diseases. In this study, we present the clinical and genetic data from a three-generation family (grandfather–mother–son) harboring a novel non-canonical splice site variant in the *PHEX* gene.

Splicing variants account for 17–22% of all reported likely pathogenic variants in the *PHEX* gene: 103 out of 620 reported variants in the Human Gene Mutation Database (https://www.hgmd.cf.ac.uk/ac/all.php, accessed on 1 March 2024); 118 out of 535 reported variants in Simple ClinVar (https://simple-clinvar.broadinstitute.org; accessed on 1 March 2024). Most variants altering the canonical ±1 or 2 splice sites (donor GT and acceptor AG) are classified as (likely) pathogenic variants since they result in aberrant splicing from pre-mRNA transcripts into mature mRNA. Conversely, non-canonical splice site variants, occurring surrounding the ±1 or 2 splice sites, require experimental evidence of splicing alteration, by either RNA or protein analysis, to support their classification as likely pathogenic [8].

Previous studies have investigated the effect on pre-mRNA splicing and the genotype–phenotype correlation of non-canonical splice site variants in the *PHEX* gene [9]. BinEssa et al. [10] performed an in vitro study using *PHEX* mini-genes of 14 previously reported non-canonical splice site variants. Most of these variants resulted in aberrant splicing and, interestingly, two of them (c.436+6T>C and c.1586+6T>C) resulted in the production of both wild-type and aberrant transcripts, which could lead to a milder phenotype. In fact, the first variant (c.436+6T>C) had been described in a family with three cases (mother and her two sons) who had very mild clinical manifestations of XLH without receiving any treatment [11]. In contrast, despite the prediction of a mild phenotype in the functional study, the second variant (c.1586+6T>C) was reported in a family with a severe phenotype presenting with rickets and impaired growth since infancy, bone deformities and pain, hearing loss and reduced quality in adulthood, requiring treatment with burosumab [12]. However, in this family, genetic testing was not performed by the NGS of a broad gene panel but by targeted Sanger sequencing only of the *PHEX* gene. This fact might allow us to speculate that an additional pathogenic variant in another gene involved in renal phosphate wasting might contribute to the severe phenotype of this family.

In our case, the RNA study showed that the non-canonical *PHEX* c.1900-4_1900-1dup variant gave rise to both wild-type and aberrant-spliced *PHEX* transcripts in all three family members (Figure 1). The presence of a significant proportion of wild-type transcripts in both hemizygous males indicates that the spliceosome machinery recognizes both the authentic and the mutated splice sites, which may explain their mild XLH phenotype. Furthermore, the complete absence of XLH manifestations in the mother could potentially be attributed to a skewed X chromosome inactivation, which may favor the expression of the wild-type *PHEX* transcript [13]. This case supports a genotype–phenotype correlation due to haploinsufficiency dosage sensitivity. It also allows us to hypothesize that the discrepancy between the estimated prevalence of XLH of 1/20,000 individuals and the lower prevalence frequently observed in large population groups could be due to an underdiagnosis of the mildest cases [14].

The high phenotypic variability in XLH, as well as of many other monogenic diseases, may be due to various factors, including allelic heterogeneity, mosaicism, X chromosome inactivation, splicing, incomplete penetrance, oligogenic inheritance, epigenetic regulation, modifier genes and environmental factors [15,16]. Mild XLH phenotypes have been described in patients carrying variants located in non-coding regions of the *PHEX* gene, such as the regulatory variant c.*231A>G in the 3′-UTR region [17], as well as in a patient carrying a mosaic variant [18]. Our case could be considered an example of phenotypic variability in XLH due to a non-canonical splice variant. We could speculate that the child’s prognosis will likely be a very mild form of XLH, similar to that observed in his grandfather. However, we must approach the genetic counseling of this family with extreme caution, as we have been unable to assess the percentage of wild type and mutated transcripts in the disease target organs.

In conclusion, this case illustrates the significant phenotypic variability observed in XLH and highlights the critical role of genetic testing in diagnosing cases with very mild manifestations of the disease. Our findings contribute to the understanding of how splicing variants influence the wide clinical spectrum of XLH, thereby enriching our understanding of its variable expressivity.

## Figures and Tables

**Figure 1 genes-15-00679-f001:**
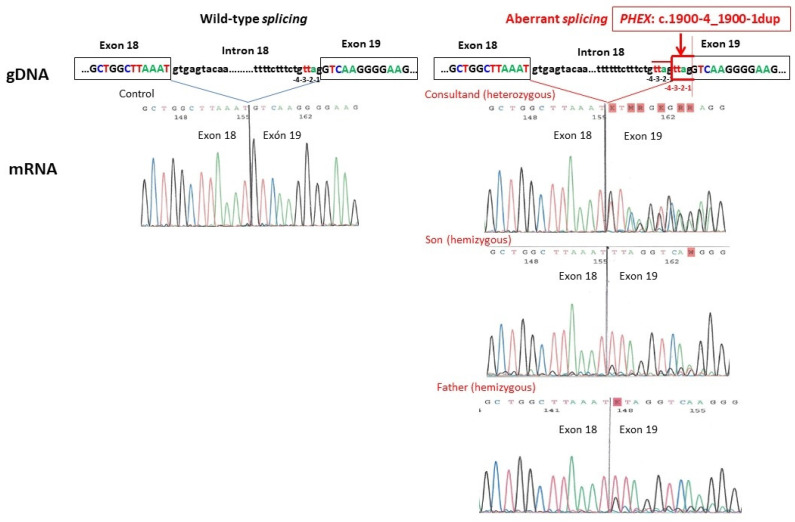
RNA analysis in all three family members of the splicing alteration caused by the variant *PHEX* (NM_000444.6): c.1900-4_1900-1dup at the genomic DNA (gDNA) level and Sanger sequences at the messenger RNA (mRNA) level compared to normal splicing. The aberrantly spliced transcript includes the 4 intronic (TTAG) duplicated nucleotides in the coding sequence.

**Table 1 genes-15-00679-t001:** Biochemical parameters in the three family members.

Patient (Age)	Serum Phosphorus (mmol/L)	Serum Calcium (mmol/L)	Serum Alkaline Phosphatase (U/L)	Serum Creatinine (µmol/L)	Serum 25OHD (nmol/L)	Serum 1,25 (OH)_2_D (ng/L)	Serum PTH (pmol/L)	FGF23 (RU/mL)	TmP/GFR (mmol/L)
Consultand (39)	1.03 (0.80–1.30)	2.54 (2.10–2.55)	NA	61 (45–80)	**33** (>50)	35 (20–54)	4.1 (1.6–6.9)	83 (≤145)	1.10 (0.84–1.23)
Father (70)	**0.69** (0.80–1.30)	2.15 (2.10–2.55)	108 (40–130)	80 (65–110)	**45** (>50)	**12** (20–54)	**9.0** (1.6–6.9)	56 (≤145)	**0.44** (0.84–1.23)
Son (3)	**1.15** (1.38–2.19)	2.47 (2.29–2.63)	262 (156–369)	41 (34–49)	73.9 (>50)	58 (20–54)	1.9 (1.6–6.9)	148 (≤230)	**0.99** (1.05–1.78)

Age-specific reference values are indicated in parentheses for each biochemical parameter. Abnormal values are highlighted in bold. FGF23: fibroblast growth factor 23; NA: not available; PTH: parathyroid hormone; TmP/GFR: maximum rate of renal tubular reabsorption of phosphate per glomerular filtration rate.

## Data Availability

Data available on request from the authors.

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
