# Peer review of "A Mild Presentation of X-Linked Hypophosphatemia Caused by a Non-Canonical Splice Site Variant in the PHEX Gene"

_genes, 2024, doi:10.3390/genes15060679_

Round 1
Reviewer 1 Report
Comments and Suggestions for Authors
This authors describes a mild case of XLH in a three-generation family, where the diagnosis was unexpectedly confirmed in a 39-year-old woman with genetic testing for a previously ill-defined tubulopathy. Novel non-canonical splice site variants in intron 18 of the PHEX gene were identified using next-generation sequencing. Analysis revealed that both the woman's father and son also carried this variant, with both abnormal and normal splicing of the PHEX gene. This well-composed case highlights how such genetic variations contribute to the variability of XLH symptoms and emphasizes the significance of genetic testing in understanding the disease. Even within this family, the authors describe different phenotypes, with the consultand’s father displaying features typical of XLH (hypophosphatemia, diminished TmP/GFR, elevated PTH and suppressed 1,25D.
Major comments:
1. There are no major concerns with this manuscript. The authors do state in the discussion that the variability in the haploinsufficient presentation (unaffected female vs. males) of the normal allele and variant c.436+6T>C and c.158_6T>C as possibly due to skewed X-chromosome inactivation. While this is a tempting possibility, given the small n-value, other possibilities exist including differences in maternal penetrance, epigenetic factors, etc.
2. The author’s analysis of the novel (absence in genome aggregation database and literature) splicing variant demonstrates that this variant is rare, not previously documented, and predicted to have a significant impact on gene splicing (by computational algorithms, SpliceAI and SQUIRLS). The authors state that these splicing variations could potentially lead to the phenotypic differences shown in Table 1. Perhaps the authors can discuss as future studies (or collaboration) the creation of a murine model to perform further functional studies to understand the biological implications that would inform not only XLH, but to validate splicing prediction models using SpliceAI and SQUIRLS.
Minor comments:
None
Reviewer 2 Report
Comments and Suggestions for Authors
The paper reports a mild form of hypophosphatemic rickets in family members with a splice site variant in the PHEX gene. Some comments may be proposed to the authors.
1. The article reports a family with members who were carriers of a mutation at PHEX gene. Of the 4 reported members only the father had a phenotype characteristic of hyphoposphatemic rickets. Thus, the father could be the “proband” of the study and not his daughter (the “consultand”) who is substantially healthy.
2. It is unclear which members underwent genetic test.
3. Patients with XLH have a low tubular reabsorption of phosphate and this leads to hypophosphatemia; the consequent lower filtered load of phosphate leads to a normal phosphate excretion in 24 hours. Therefore, from a strict quantitative point of view, hyperphosphaturia is not a characteristic of these patients. The sentence at the line 42 of page 1 should be revised.
4. The word “consultand” does not appear to me the most appropriate word to define the proband of the study.
5. It is superfluous to report Fractional Excretion of phosphate and Tubular Reabsorption of Phosphate at the line 108, as the former is complementary of the latter.
6. Excretion fraction and Tubular Reabsorption of Phosphate have FE and TRP as acronymous.
7. The son reported at the lines 110-115 should be more clearly identified between the two sons in table 1.
8. The father could be described with more details: it is unclear its clinical diagnosis, the age of development of bone deformities and the time of therapy starting should be reported.
9. Age of proband’s sons could be more clearly reported in table 1 as 2.4 could mean 2 years and 4 months (in twelfths), but it could also mean 2 year and 5 months (in tenths), (the same for 3.4 in the other son).
10. Serum phosphate was expressed as mmol/l; therefore also TmP/GFR should be expressed as mmol/l in table 1.
11. Unit to express body size should be corrected at line113.
Round 2
Reviewer 2 Report
Comments and Suggestions for Authors
No other comments by me. The Authors explained the different definitions in their manuscript and positively modified it according to reviewers' suggestions.